# DIRECTIONAL MESSAGE PASSING FOR MOLECULAR GRAPHS

**Johannes Gasteiger, Janek Groß & Stephan Günnemann**
Technical University of Munich, Germany
`{j.gasteiger,grossja,guennemann}@in.tum.de`

## ABSTRACT

Graph neural networks have recently achieved great successes in predicting quantum mechanical properties of molecules. These models represent a molecule as a graph using only the distance between atoms (nodes). They do not, however, consider the spatial direction from one atom to another, despite directional information playing a central role in empirical potentials for molecules, e.g. in angular potentials. To alleviate this limitation we propose directional message passing, in which we embed the messages passed between atoms instead of the atoms themselves. Each message is associated with a direction in coordinate space. These directional message embeddings are rotationally equivariant since the associated directions rotate with the molecule. We propose a message passing scheme analogous to belief propagation, which uses the directional information by transforming messages based on the angle between them. Additionally, we use spherical Bessel functions and spherical harmonics to construct theoretically well-founded, orthogonal representations that achieve better performance than the currently prevalent Gaussian radial basis representations while using fewer than ¼ of the parameters. We leverage these innovations to construct the directional message passing neural network (DimeNet). DimeNet outperforms previous GNNs on average by $76\%$ on MD17 and by $31\%$ on QM9. Our implementation is available online. [1]

## 1 INTRODUCTION

In recent years scientists have started leveraging machine learning to reduce the computation time required for predicting molecular properties from a matter of hours and days to mere milliseconds. With the advent of graph neural networks (GNNs) this approach has recently experienced a small revolution, since they do not require any form of manual feature engineering and significantly outperform previous models (Gilmer et al., 2017; Schütt et al., 2017). GNNs model the complex interactions between atoms by embedding each atom in a high-dimensional space and updating these embeddings by passing messages between atoms. By predicting the potential energy these models effectively learn an empirical potential function. Classically, these functions have been modeled as the sum of four parts: (Leach, 2001)

$$E = E_{\text{bonds}} + E_{\text{angle}} + E_{\text{torsion}} + E_{\text{non-bonded}}, \tag{1}$$

where $E_{\text{bonds}}$ models the dependency on bond lengths, $E_{\text{angle}}$ on the angles between bonds, $E_{\text{torsion}}$ on bond rotations, i.e. the dihedral angle between two planes defined by pairs of bonds, and $E_{\text{non-bonded}}$ models interactions between unconnected atoms, e.g. via electrostatic or van der Waals interactions. The update messages in GNNs, however, only depend on the previous atom embeddings and the pairwise distances between atoms – not on directional information such as bond angles and rotations. Thus, GNNs lack the second and third terms of this equation and can only model them via complex higher-order interactions of messages. Extending GNNs to model them directly is not straightforward since GNNs solely rely on pairwise distances, which ensures their invariance to translation, rotation, and inversion of the molecule, which are important physical requirements.

In this paper, we propose to resolve this restriction by using embeddings associated with the directions to neighboring atoms, i.e. by embedding atoms as a set of messages. These directional message

---

[1] `https://www.daml.in.tum.de/dimenet`

embeddings are equivariant with respect to the above transformations since the directions move *with* the molecule. Hence, they preserve the relative directional information between neighboring atoms. We propose to let message embeddings interact based on the distance between atoms and the angle between directions. Both distances and angles are invariant to translation, rotation, and inversion of the molecule, as required. Additionally, we show that the distance and angle can be jointly represented in a principled and effective manner by using spherical Bessel functions and spherical harmonics. We leverage these innovations to construct the directional message passing neural network (DimeNet). DimeNet can learn both molecular properties and atomic forces. It is twice continuously differentiable and solely based on the atom types and coordinates, which are essential properties for performing molecular dynamics simulations. DimeNet outperforms previous GNNs on average by $76\,\%$ on MD17 and by $31\,\%$ on QM9. Our paper's main contributions are:

1. Directional message passing, which allows GNNs to incorporate directional information by connecting recent advances in the fields of equivariance and graph neural networks as well as ideas from belief propagation and empirical potential functions such as Eq. 1.
2. Theoretically principled orthogonal basis representations based on spherical Bessel functions and spherical harmonics. Bessel functions achieve better performance than Gaussian radial basis functions while reducing the radial basis dimensionality by 4x or more.
3. The Directional Message Passing Neural Network (DimeNet): A novel GNN that leverages these innovations to set the new state of the art for molecular predictions and is suitable both for predicting molecular properties and for molecular dynamics simulations.

## 2 RELATED WORK

**ML for molecules.** The classical way of using machine learning for predicting molecular properties is combining an expressive, hand-crafted representation of the atomic neighborhood (Bartók et al., 2013) with Gaussian processes (Bartók et al., 2010; 2017; Chmiela et al., 2017) or neural networks (Behler & Parrinello, 2007). Recently, these methods have largely been superseded by graph neural networks, which do not require any hand-crafted features but learn representations solely based on the atom types and coordinates molecules (Duvenaud et al., 2015; Gilmer et al., 2017; Schütt et al., 2017; Hy et al., 2018; Unke & Meuwly, 2019). Our proposed message embeddings can also be interpreted as directed edge embeddings or embeddings on the line graph (Chen et al., 2019b). (Undirected) edge embeddings have already been used in previous GNNs for molecules (Jørgensen et al., 2018; Chen et al., 2019a). However, these GNNs use both node and edge embeddings and do not leverage any directional information.

**Graph neural networks.** GNNs were first proposed in the 90s (Baskin et al., 1997; Sperduti & Starita, 1997) and 00s (Gori et al., 2005; Scarselli et al., 2009). General GNNs have been largely inspired by their application to molecular graphs and have started to achieve breakthrough performance in various tasks at around the same time the molecular variants did (Kipf & Welling, 2017; Gasteiger et al., 2019; Zambaldi et al., 2019). Some recent progress has been focused on GNNs that are more powerful than the 1-Weisfeiler-Lehman test of isomorphism (Morris et al., 2019; Maron et al., 2019). However, for molecular predictions these models are significantly outperformed by GNNs focused on molecules (see Sec. 7). Some recent GNNs have incorporated directional information by considering the change in local coordinate systems per atom (Ingraham et al., 2019). However, this approach breaks permutation invariance and is therefore only applicable to chain-like molecules (e.g. proteins).

**Equivariant neural networks.** Group equivariance as a principle of modern machine learning was first proposed by Cohen & Welling (2016). Following work has generalized this principle to spheres (Cohen et al., 2018), molecules (Thomas et al., 2018), volumetric data (Weiler et al., 2018), and general manifolds (Cohen et al., 2019). Equivariance with respect to continuous rotations has been achieved so far by switching back and forth between Fourier and coordinate space in each layer (Cohen et al., 2018) or by using a fully Fourier space model (Kondor et al., 2018; Anderson et al., 2019). The former introduces major computational overhead and the latter imposes significant constraints on model construction, such as the inability of using non-linearities. Our proposed solution does not suffer from either of those limitations.

## 3 REQUIREMENTS FOR MOLECULAR PREDICTIONS

In recent years machine learning has been used to predict a wide variety of molecular properties, both low-level quantum mechanical properties such as potential energy, energy of the highest occupied molecular orbital (HOMO), and the dipole moment and high-level properties such as toxicity, permeability, and adverse drug reactions (Wu et al., 2018). In this work we will focus on scalar regression targets, i.e. targets $t \in \mathbb{R}$. A molecule is uniquely defined by the atomic numbers $\boldsymbol{z} = \{z_1, \ldots, z_N\}$ and positions $\boldsymbol{X} = \{\boldsymbol{x}_1, \ldots, \boldsymbol{x}_N\}$. Some models additionally use auxiliary information $\boldsymbol{\Theta}$ such as bond types or electronegativity of the atoms. We do not include auxiliary features in this work since they are hand-engineered and non-essential. In summary, we define an ML model for molecular prediction with parameters $\theta$ via $f_\theta : \{\boldsymbol{X}, \boldsymbol{z}\} \to \mathbb{R}$.

**Symmetries and invariances.** All molecular predictions must obey some basic laws of physics, either explicitly or implicitly. One important example of such are the fundamental symmetries of physics and their associated invariances. In principle, these invariances can be learned by any neural network via corresponding weight matrix symmetries (Ravanbakhsh et al., 2017). However, not explicitly incorporating them into the model introduces duplicate weights and increases training time and complexity. The most essential symmetries are translational and rotational invariance (follows from homogeneity and isotropy), permutation invariance (follows from the indistinguishability of particles), and symmetry under parity, i.e. under sign flips of single spatial coordinates.

**Molecular dynamics.** Additional requirements arise when the model should be suitable for molecular dynamics (MD) simulations and predict the forces $\boldsymbol{F}_i$ acting on each atom. The force field is a conservative vector field since it must satisfy conservation of energy (the necessity of which follows from homogeneity of time (Noether, 1918)). The easiest way of defining a conservative vector field is via the gradient of a potential function. We can leverage this fact by predicting a potential instead of the forces and then obtaining the forces via backpropagation to the atom coordinates, i.e. $\boldsymbol{F}_i(\boldsymbol{X}, \boldsymbol{z}) = -\frac{\partial}{\partial \boldsymbol{x}_i} f_\theta(\boldsymbol{X}, \boldsymbol{z})$. We can even directly incorporate the forces in the training loss and directly train a model for MD simulations (Pukrittayakamee et al., 2009):

$$\mathcal{L}_{\text{MD}}(\boldsymbol{X}, \boldsymbol{z}) = \left| f_\theta(\boldsymbol{X}, \boldsymbol{z}) - \hat{t}(\boldsymbol{X}, \boldsymbol{z}) \right| + \frac{\rho}{3N} \sum_{i=1}^{N} \sum_{\alpha=1}^{3} \left| -\frac{\partial f_\theta(\boldsymbol{X}, \boldsymbol{z})}{\partial \boldsymbol{x}_{i\alpha}} - \hat{F}_{i\alpha}(\boldsymbol{X}, \boldsymbol{z}) \right|, \qquad (2)$$

where the target $\hat{t} = \hat{E}$ is the ground-truth energy (usually available as well), $\hat{\boldsymbol{F}}$ are the ground-truth forces, and the hyperparameter $\rho$ sets the forces' loss weight. For stable simulations $\boldsymbol{F}_i$ must be continuously differentiable and the model $f_\theta$ itself therefore twice continuously differentiable. We hence cannot use discontinuous transformations such as ReLU non-linearities. Furthermore, since the atom positions $\boldsymbol{X}$ can change arbitrarily we cannot use pre-computed auxiliary information $\boldsymbol{\Theta}$ such as bond types.

## 4 DIRECTIONAL MESSAGE PASSING

**Graph neural networks.** Graph neural networks treat the molecule as a graph, in which the nodes are atoms and edges are defined either via a predefined molecular graph or simply by connecting atoms that lie within a cutoff distance $c$. Each edge is associated with a pairwise distance between atoms $d_{ij} = \|\boldsymbol{x}_i - \boldsymbol{x}_j\|_2$. GNNs implement all of the above physical invariances by construction since they only use pairwise distances and not the full atom coordinates. However, note that a predefined molecular graph or a step function-like cutoff cannot be used for MD simulations since this would introduce discontinuities in the energy landscape. GNNs represent each atom $i$ via an atom embedding $\boldsymbol{h}_i \in \mathbb{R}^H$. The atom embeddings are updated in each layer by passing messages along the molecular edges. Messages are usually transformed based on an edge embedding $\boldsymbol{e}_{(ij)} \in \mathbb{R}^{H_e}$ and summed over the atom's neighbors $\mathcal{N}_i$, i.e. the embeddings are updated in layer $l$ via

$$\boldsymbol{h}_i^{(l+1)} = f_{\text{update}}(\boldsymbol{h}_i^{(l)}, \sum_{j \in \mathcal{N}_i} f_{\text{int}}(\boldsymbol{h}_j^{(l)}, \boldsymbol{e}_{(ij)}^{(l)})), \qquad (3)$$

with the update function $f_{\text{update}}$ and the interaction function $f_{\text{int}}$, which are both commonly implemented using neural networks. The edge embeddings $\boldsymbol{e}_{(ij)}^{(l)}$ usually only depend on the interatomic distances, but can also incorporate additional bond information (Gilmer et al., 2017) or be recursively updated in each layer using the neighboring atom embeddings (Jørgensen et al., 2018).

**Directionality.** In principle, the pairwise distance matrix contains the full geometrical information of the molecule. However, GNNs do not use the full distance matrix since this would mean passing messages globally between all pairs of atoms, which increases computational complexity and can lead to overfitting. Instead, they usually use a cutoff distance $c$, which means they cannot distinguish between certain molecules (Xu et al., 2019). E.g. at a cutoff of roughly 2 Å a regular GNN would not be able to distinguish between a hexagonal (e.g. Cyclohexane) and two triangular molecules (e.g. Cyclopropane) with the same bond lengths since the neighborhoods of each atom are exactly the same for both (see Appendix, Fig. 6). This problem can be solved by modeling the directions to neighboring atoms instead of just their distances. A principled way of doing so while staying invariant to a transformation group $G$ (such as described in Sec. 3) is via group-equivariance (Cohen & Welling, 2016). A function $f : X \to Y$ is defined as being equivariant if $f(\varphi_g^X(x)) = \varphi_g^Y(f(x))$, with the group action in the input and output space $\varphi_g^X$ and $\varphi_g^Y$. However, equivariant CNNs only achieve equivariance with respect to a discrete set of rotations (Cohen & Welling, 2016). For a precise prediction of molecular properties we need *continuous* equivariance with respect to rotations, i.e. to the SO(3) group.

**Directional embeddings.** We solve this problem by noting that an atom by itself is rotationally invariant. This invariance is only broken by neighboring atoms that interact with it, i.e. those inside the cutoff $c$. Since each neighbor breaks up to one rotational invariance they also introduce additional degrees of freedom, which we need to represent in our model. We can do so by generating a separate embedding $\boldsymbol{m}_{ji}$ for each atom $i$ and neighbor $j$ by applying the same learned filter in the direction of each neighboring atom (in contrast to equivariant CNNs, which apply filters in fixed, global directions). These directional embeddings are equivariant with respect to global rotations since the associated directions rotate *with* the molecule and hence conserve the relative directional information between neighbors.

**Representation via joint 2D basis.** We use the directional information associated with each embedding by leveraging the angle $\alpha_{(kj,ji)} = \angle \boldsymbol{x}_k \boldsymbol{x}_j \boldsymbol{x}_i$ when aggregating the neighboring embeddings $\boldsymbol{m}_{kj}$ of $\boldsymbol{m}_{ji}$. We combine the angle with the interatomic distance $d_{kj}$ associated with the incoming message $\boldsymbol{m}_{kj}$ and jointly represent both in $\boldsymbol{a}_{\text{SBF}}^{(kj,ji)} \in \mathbb{R}^{N_{\text{SHBF}} \cdot N_{\text{SRBF}}}$ using a 2D representation based on spherical Bessel functions and spherical harmonics, as explained in Sec. 5. We empirically found that this basis representation provides a better inductive bias than the raw angle alone. Note that by only using interatomic distances and angles our model becomes invariant to rotations.

**Message embeddings.** The directional embedding $\boldsymbol{m}_{ji}$ associated with the atom pair $ji$ can be thought of as a message being sent from atom $j$ to atom $i$. Hence, in analogy to belief propagation, we embed each atom $i$ using a set of incoming messages $\boldsymbol{m}_{ji}$, i.e. $\boldsymbol{h}_i = \sum_{j \in \mathcal{N}_i} \boldsymbol{m}_{ji}$, and update the message $\boldsymbol{m}_{ji}$ based on the incoming messages $\boldsymbol{m}_{kj}$ (Yedidia et al., 2003). Hence, as illustrated in Fig. 1, we define the update function and aggregation scheme for message embeddings as

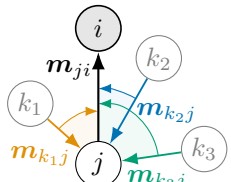

Figure 1: Aggregation scheme for message embeddings.

$$\boldsymbol{m}_{ji}^{(l+1)} = f_{\text{update}}(\boldsymbol{m}_{ji}^{(l)}, \sum_{k \in \mathcal{N}_j \setminus \{i\}} f_{\text{int}}(\boldsymbol{m}_{kj}^{(l)}, \boldsymbol{e}_{\text{RBF}}^{(ji)}, \boldsymbol{a}_{\text{SBF}}^{(kj,ji)})), \quad (4)$$

where $\boldsymbol{e}_{\text{RBF}}^{(ji)}$ denotes the radial basis function representation of the interatomic distance $d_{ji}$, which will be discussed in Sec. 5. We found this aggregation scheme to not only have a nice analogy to belief propagation, but also to empirically perform better than alternatives. Note that since $f_{\text{int}}$ now incorporates the angle between atom pairs, or bonds, we have enabled our model to directly learn the angular potential $E_{\text{angle}}$, the second term in Eq. 1. Moreover, the message embeddings are essentially embeddings of atom pairs, as used by the provably more powerful GNNs based on higher-order Weisfeiler-Lehman tests of isomorphism. Our model can therefore provably distinguish molecules that a regular GNN cannot (e.g. the previous example of a hexagonal and two triangular molecules) (Morris et al., 2019).

## 5 PHYSICALLY BASED REPRESENTATIONS

**Representing distances and angles.** For the interaction function $f_{\text{int}}$ in Eq. 4 we use a joint representation $\boldsymbol{a}_{\text{SBF}}^{(kj,ji)}$ of the angles $\alpha_{(kj,ji)}$ between message embeddings and the interatomic

distances $d_{kj} = \|\boldsymbol{x}_k - \boldsymbol{x}_j\|_2$, as well as a representation $\boldsymbol{e}_{\text{RBF}}^{(ji)}$ of the distances $d_{ji}$. Earlier works have used a set of Gaussian radial basis functions to represent interatomic distances, with tightly spaced means that are distributed e.g. uniformly (Schütt et al., 2017) or exponentially (Unke & Meuwly, 2019). Similar in spirit to the functional bases used by steerable CNNs (Cohen & Welling, 2017; Cheng et al., 2019) we propose to use an orthogonal basis instead, which reduces redundancy and thus improves parameter efficiency. Furthermore, a basis chosen according to the properties of the modeled system can even provide a helpful inductive bias. We therefore derive a proper basis representation for quantum systems next.

**From Schrödinger to Fourier-Bessel.** To construct a basis representation in a principled manner we first consider the space of possible solutions. Our model aims at approximating results of density functional theory (DFT) calculations, i.e. results given by an electron density $\langle \Psi(\boldsymbol{d}) | \Psi(\boldsymbol{d}) \rangle$, with the electron wave function $\Psi(\boldsymbol{d})$ and $\boldsymbol{d} = \boldsymbol{x}_k - \boldsymbol{x}_j$. The solution space of $\Psi(\boldsymbol{d})$ is defined by the time-independent Schrödinger equation $\left( -\frac{\hbar^2}{2m} \nabla^2 + V(\boldsymbol{d}) \right) \Psi(\boldsymbol{d}) = E\Psi(\boldsymbol{d})$, with constant mass $m$ and energy $E$. We do not know the potential $V(\boldsymbol{d})$ and so choose it in an uninformative way by simply setting it to 0 inside the cutoff distance $c$ (up to which we pass messages between atoms) and to $\infty$ outside. Hence, we arrive at the Helmholtz equation $(\nabla^2 + k^2)\Psi(\boldsymbol{d}) = 0$, with the wave number $k = \frac{\sqrt{2mE}}{\hbar}$ and the boundary condition $\Psi(c) = 0$ at the cutoff $c$. Separation of variables in polar coordinates $(d, \alpha, \varphi)$ yields the solution (Griffiths & Schroeter, 2018)

$$\Psi(d, \alpha, \varphi) = \sum_{l=0}^{\infty} \sum_{m=-l}^{l} (a_{lm} j_l(kd) + b_{lm} y_l(kd)) Y_l^m(\alpha, \varphi), \tag{5}$$

with the spherical Bessel functions of the first and second kind $j_l$ and $y_l$ and the spherical harmonics $Y_l^m$. As common in physics we only use the regular solutions, i.e. those that do not approach $-\infty$ at the origin, and hence set $b_{lm} = 0$. Recall that our first goal is to construct a joint 2D basis for $d_{kj}$ and $\alpha_{(kj,ji)}$, i.e. a function that depends on $d$ and a single angle $\alpha$. To achieve this we set $m = 0$ and obtain $\Psi_{\text{SBF}}(d, \alpha) = \sum_l a_l j_l(kd) Y_l^0(\alpha)$. The boundary conditions are satisfied by setting $k = \frac{z_{ln}}{c}$, where $z_{ln}$ is the $n$-th root of the $l$-order Bessel function, which are precomputed numerically. Normalizing $\Psi_{\text{SBF}}$ inside the cutoff distance $c$ yields the 2D spherical Fourier-Bessel basis $\tilde{\boldsymbol{a}}_{\text{SBF}}^{(kj,ji)} \in \mathbb{R}^{N_{\text{SHBF}} \cdot N_{\text{SRBF}}}$, which is illustrated in Fig. 2 and defined by



$$\tilde{a}_{\text{SBF},ln}(d, \alpha) = \sqrt{\frac{2}{c^3 j_{l+1}^2(z_{ln})}} j_l(\frac{z_{ln}}{c} d) Y_l^0(\alpha), \tag{6}$$

with $l \in [0 \,.\, . \, N_{\text{SHBF}} - 1]$ and $n \in [1 \,.\, . \, N_{\text{SRBF}}]$. Our second goal is constructing a radial basis for $d_{ji}$, i.e. a function that solely depends on $d$ and not on the angles $\alpha$ and $\varphi$. We achieve this by setting $l = m = 0$ and obtain $\Psi_{\text{RBF}}(d) = a j_0(\frac{z_{0,n}}{c} d)$, with roots at $z_{0,n} = n\pi$. Normalizing this function on $[0, c]$ and using $j_0(d) = \sin(d)/d$ gives the radial basis $\tilde{\boldsymbol{e}}_{\text{RBF}} \in \mathbb{R}^{N_{\text{RBF}}}$, as shown in Fig. 3 and defined by

Figure 2: 2D spherical Fourier-Bessel basis $\tilde{a}_{\text{SBF},ln}(d, \alpha)$.

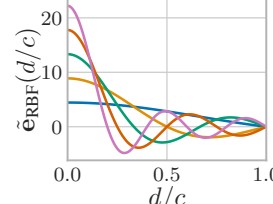

$$\tilde{e}_{\text{RBF},n}(d) = \sqrt{\frac{2}{c}} \frac{\sin(\frac{n\pi}{c} d)}{d}, \tag{7}$$

with $n \in [1 \,.\, . \, N_{\text{RBF}}]$. Both of these bases are purely real-valued and orthogonal in the domain of interest. They furthermore enable us to

Figure 3: Radial Bessel basis for $N_{\text{RBF}} = 5$.

bound the highest-frequency components by $\omega_\alpha \leq \frac{N_{\text{SHBF}}}{2\pi}$, $\omega_{d_{kj}} \leq \frac{N_{\text{SRBF}}}{c}$, and $\omega_{d_{ji}} \leq \frac{N_{\text{RBF}}}{c}$. This restriction is an effective way of regularizing the model and ensures that predictions are stable to small perturbations. We found $N_{\text{SRBF}} = 6$ and $N_{\text{RBF}} = 16$ radial basis functions to be more than sufficient. Note that $N_{\text{RBF}}$ is 4x lower than PhysNet's 64 (Unke & Meuwly, 2019) and 20x lower than SchNet's 300 radial basis functions (Schütt et al., 2017).

**Continuous cutoff.** $\tilde{\boldsymbol{a}}_{\text{SBF}}^{(kj,ji)}$ and $\tilde{\boldsymbol{e}}_{\text{RBF}}(d)$ are not twice continuously differentiable due to the step function cutoff at $c$. To alleviate this problem we introduce an envelope function $u(d)$ that has a root of multiplicity 3 at $d = c$, causing the final functions $\boldsymbol{a}_{\text{RBF}}(d) = u(d)\tilde{\boldsymbol{a}}_{\text{RBF}}(d)$ and $\boldsymbol{e}_{\text{RBF}}(d) =$

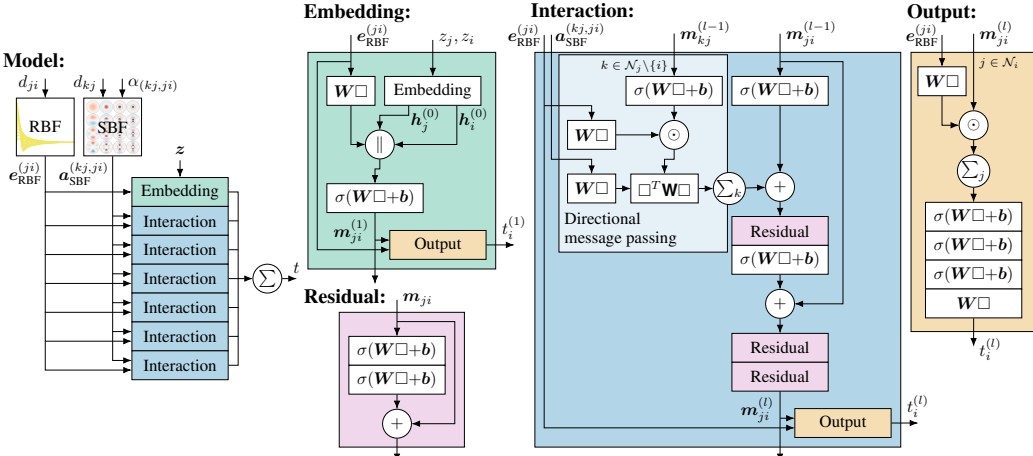

Figure 4: The DimeNet architecture. $\square$ denotes the layer's input and $\|$ denotes concatenation. The distances $d_{ji}$ are represented using spherical Bessel functions and the distances $d_{kj}$ and angles $\alpha_{(kj,ji)}$ are jointly represented using a 2D spherical Fourier-Bessel basis. An embedding block generates the inital message embeddings $\boldsymbol{m}_{ji}$. These embeddings are updated in multiple interaction blocks via directional message passing, which uses the neighboring messages $\boldsymbol{m}_{kj}, k \in \mathcal{N}_j \setminus \{i\}$, the 2D representations $\boldsymbol{a}_{\text{SBF}}^{(kj,ji)}$, and the distance representations $\boldsymbol{e}_{\text{RBF}}^{(ji)}$. Each block passes the resulting embeddings to an output block, which transforms them using the radial basis $\boldsymbol{e}_{\text{RBF}}^{(ji)}$ and sums them up per atom. Finally, the outputs of all layers are summed up to generate the prediction.

$u(d)\tilde{\boldsymbol{e}}_{\text{RBF}}(d)$ and their first and second derivatives to go to 0 at the cutoff. We achieve this with the polynomial

$$u(d) = 1 - \frac{(p+1)(p+2)}{2}d^p + p(p+2)d^{p+1} - \frac{p(p+1)}{2}d^{p+2}, \tag{8}$$

where $p \in \mathbb{N}_0$. We did not find the model to be sensitive to different choices of envelope functions and choose $p = 6$. Note that using an envelope function causes the bases to lose their orthonormality, which we did not find to be a problem in practice. We furthermore fine-tune the Bessel wave numbers $k_n = \frac{n\pi}{c}$ used in $\tilde{\boldsymbol{e}}_{\text{RBF}} \in \mathbb{R}^{N_{\text{RBF}}}$ via backpropagation after initializing them to these values, which we found to give a small boost in prediction accuracy.

## 6 DIRECTIONAL MESSAGE PASSING NEURAL NETWORK (DIMENET)

The Directional Message Passing Neural Network's (DimeNet) design is based on a streamlined version of the PhysNet architecture (Unke & Meuwly, 2019), in which we have integrated directional message passing and spherical Fourier-Bessel representations. DimeNet generates predictions that are invariant to atom permutations and translation, rotation and inversion of the molecule. DimeNet is suitable both for the prediction of various molecular properties and for molecular dynamics (MD) simulations. It is twice continuously differentiable and able to learn and predict atomic forces via backpropagation, as described in Sec. 3. The predicted forces fulfill energy conservation by construction and are equivariant with respect to permutation and rotation. Model differentiability in combination with basis representations that have bounded maximum frequencies furthermore guarantees smooth predictions that are stable to small deformations. Fig. 4 gives an overview of the architecture.

**Embedding block.** Atomic numbers are represented by learnable, randomly initialized atom type embeddings $\boldsymbol{h}_i^{(0)} \in \mathbb{R}^F$ that are shared across molecules. The first layer generates message embeddings from these and the distance between atoms via

$$\boldsymbol{m}_{ji}^{(1)} = \sigma([\boldsymbol{h}_j^{(0)} \| \boldsymbol{h}_i^{(0)} \| \boldsymbol{e}_{\text{RBF}}^{(ji)}] \boldsymbol{W} + \boldsymbol{b}), \tag{9}$$

where $\|$ denotes concatenation and the weight matrix $\boldsymbol{W}$ and bias $\boldsymbol{b}$ are learnable.

Table 1: MAE on QM9. DimeNet sets the state of the art on 11 targets, outperforming the second-best model on average by 31 % (mean std. MAE).

| Target | Unit | PPGN | SchNet | PhysNet | MEGNet-s | Cormorant | **DimeNet** |
|---|---|---|---|---|---|---|---|
| $\mu$ | D | 0.047 | 0.033 | 0.0529 | 0.05 | 0.13 | **0.0286** |
| $\alpha$ | $a_0{}^3$ | 0.131 | 0.235 | 0.0615 | 0.081 | 0.092 | **0.0469** |
| $\epsilon_{\text{HOMO}}$ | meV | 40.3 | 41 | 32.9 | 43 | 36 | **27.8** |
| $\epsilon_{\text{LUMO}}$ | meV | 32.7 | 34 | 24.7 | 44 | 36 | **19.7** |
| $\Delta\epsilon$ | meV | 60.0 | 63 | 42.5 | 66 | 60 | **34.8** |
| $\langle R^2 \rangle$ | $a_0{}^2$ | 0.592 | **0.073** | 0.765 | 0.302 | 0.673 | 0.331 |
| ZPVE | meV | 3.12 | 1.7 | 1.39 | 1.43 | 1.98 | **1.29** |
| $U_0$ | meV | 36.8 | 14 | 8.15 | 12 | 28 | **8.02** |
| $U$ | meV | 36.8 | 19 | 8.34 | 13 | - | **7.89** |
| $H$ | meV | 36.3 | 14 | 8.42 | 12 | - | **8.11** |
| $G$ | meV | 36.4 | 14 | 9.40 | 12 | - | **8.98** |
| $c_{\text{v}}$ | $\frac{\text{cal}}{\text{mol K}}$ | 0.055 | 0.033 | 0.0280 | 0.029 | 0.031 | **0.0249** |
| std. MAE | % | 1.84 | 1.76 | 1.37 | 1.80 | 2.14 | **1.05** |
| logMAE | - | $-4.64$ | $-5.17$ | $-5.35$ | $-5.17$ | $-4.75$ | **$-5.57$** |

**Interaction block.** The embedding block is followed by multiple stacked interaction blocks. This block implements $f_{\text{int}}$ and $f_{\text{update}}$ of Eq. 4 as shown in Fig. 4. Note that the 2D representation $a_{\text{SBF}}^{(kj,ji)}$ is first transformed into an $N_{\text{bilinear}}$-dimensional representation via a linear layer. The main purpose of this is to make the dimensionality of $a_{\text{SBF}}^{(kj,ji)}$ independent of the subsequent bilinear layer, which uses a comparatively large $N_{\text{bilinear}} \times F \times F$-dimensional weight tensor. We have also experimented with using a bilinear layer for the radial basis representation, but found that the element-wise multiplication $e_{\text{RBF}}^{(ji)} W \odot m_{kj}$ performs better, which suggests that the 2D representations require more complex transformations than radial information alone. The interaction block transforms each message embedding $m_{ji}$ using multiple residual blocks, which are inspired by ResNet (He et al., 2016) and consist of two stacked dense layers and a skip connection.

**Output block.** The message embeddings after each block (including the embedding block) are passed to an output block. The output block transforms each message embedding $m_{ji}$ using the radial basis $e_{\text{RBF}}^{(ji)}$, which ensures continuous differentiability and slightly improves performance. Afterwards the incoming messages are summed up per atom $i$ to obtain $h_i = \sum_j m_{ji}$, which is then transformed using multiple dense layers to generate the atom-wise output $t_i^{(l)}$. These outputs are then summed up to obtain the final prediction $t = \sum_i \sum_l t_i^{(l)}$.

**Continuous differentiability.** Multiple model choices were necessary to achieve twice continuous model differentiability. First, DimeNet uses the self-gated Swish activation function $\sigma(x) = x \cdot \text{sigmoid}(x)$ (Ramachandran et al., 2018) instead of a regular ReLU activation function. Second, we multiply the radial basis functions $\tilde{e}_{\text{RBF}}(d)$ with an envelope function $u(d)$ that has a root of multiplicity 3 at the cutoff $c$. Finally, DimeNet does not use any auxiliary data but relies on atom types and positions alone.

## 7 EXPERIMENTS

**Models.** For hyperparameter choices and training setup see Appendix B. We use 6 state-of-the-art models for comparison: SchNet (Schütt et al., 2017), PhysNet (results based on the reference implementation) (Unke & Meuwly, 2019), provably powerful graph networks (PPGN, results provided by the original authors) (Maron et al., 2019), MEGNet-simple (without auxiliary information) (Chen et al., 2019a), Cormorant (Anderson et al., 2019), and symmetrized gradient-domain machine learning (sGDML) (Chmiela et al., 2018). Note that sGDML cannot be used for QM9 since it can only be trained on a single molecule.

**QM9.** We test DimeNet's performance for predicting molecular properties using the common QM9 benchmark (Ramakrishnan et al., 2014). It consists of roughly 130 000 molecules in equilibrium

Table 2: MAE on MD17 using 1000 training samples (energies in $\frac{\text{kcal}}{\text{mol}}$, forces in $\frac{\text{kcal}}{\text{mol Å}}$). DimeNet outperforms SchNet by a large margin and performs roughly on par with sGDML.

| | | sGDML | SchNet | **DimeNet** |
|---|---|---|---|---|
| Aspirin | Energy | **0.19** | 0.37 | 0.204 |
| | Forces | 0.68 | 1.35 | **0.499** |
| Benzene | Energy | 0.10 | **0.08** | 0.078 |
| | Forces | **0.06** | 0.31 | 0.187 |
| Ethanol | Energy | 0.07 | 0.08 | **0.064** |
| | Forces | 0.33 | 0.39 | **0.230** |
| Malonaldehyde | Energy | **0.10** | 0.13 | 0.104 |
| | Forces | 0.41 | 0.66 | **0.383** |
| Naphthalene | Energy | **0.12** | 0.16 | 0.122 |
| | Forces | **0.11** | 0.58 | 0.215 |
| Salicylic acid | Energy | **0.12** | 0.20 | 0.134 |
| | Forces | **0.28** | 0.85 | 0.374 |
| Toluene | Energy | **0.10** | 0.12 | 0.102 |
| | Forces | **0.14** | 0.57 | 0.216 |
| Uracil | Energy | **0.11** | 0.14 | 0.115 |
| | Forces | **0.24** | 0.56 | 0.301 |
| std. MAE (%) | Energy | 2.53 | 3.32 | **2.49** |
| | Forces | **1.01** | 2.38 | 1.10 |

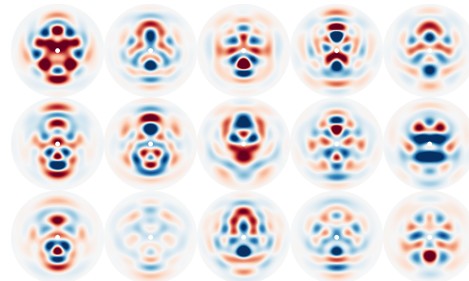

Figure 5: Examples of DimeNet filters. They exhibit a clear 2D structure. For details see Appendix D.

Table 3: Ablation studies using multi-task learning on QM9. All of our contributions have a significant impact on performance.

| Variation | $\frac{\text{MAE}}{\text{MAE DimeNet}}$ | $\Delta$logMAE |
|---|---|---|
| Gaussian RBF | 110 % | 0.10 |
| $N_{\text{SHBF}} = 1$ | 126 % | 0.11 |
| Node embeddings | 168 % | 0.45 |

with up to 9 heavy C, O, N, and F atoms. We use $110\,000$ molecules in the training, $10\,000$ in the validation and $10\,831$ in the test set. We only use the atomization energy for $U_0$, $U$, $H$, and $G$, i.e. subtract the atomic reference energies, which are constant per atom type, and perform the training using eV. In Table 1 we report the mean absolute error (MAE) of each target and the overall mean standardized MAE (std. MAE) and mean standardized logMAE (for details see Appendix C). We predict $\Delta\epsilon$ simply by taking $\epsilon_{\text{LUMO}} - \epsilon_{\text{HOMO}}$, since it is calculated in exactly this way by DFT calculations. We train a separate model for each target, which significantly improves results compared to training a single shared model for all targets (see App. E). DimeNet sets the new state of the art on 11 out of 12 targets and decreases mean std. MAE by $31\,\%$ and mean logMAE by 0.22 compared to the second-best model.

**MD17.** We use MD17 (Chmiela et al., 2017) to test model performance in molecular dynamics simulations. The goal of this benchmark is predicting both the energy and atomic forces of eight small organic molecules, given the atom coordinates of the thermalized (i.e. non-equilibrium, slightly moving) system. The ground truth data is computed via molecular dynamics simulations using DFT. A separate model is trained for each molecule, with the goal of providing highly accurate individual predictions. This dataset is commonly used with $50\,000$ training and $10\,000$ validation and test samples. We found that DimeNet can match state-of-the-art performance in this setup. E.g. for Benzene, depending on the force weight $\rho$, DimeNet achieves $0.035\,\text{kcal}\,\text{mol}^{-1}$ MAE for the energy or $0.07\,\text{kcal}\,\text{mol}^{-1}$ and $0.17\,\text{kcal}\,\text{mol}^{-1}\,\text{Å}^{-1}$ for energy and forces, matching the results reported by Anderson et al. (2019) and Unke & Meuwly (2019). However, this accuracy is two orders of magnitude below the DFT calculation's accuracy (approx. $2.3\,\text{kcal}\,\text{mol}^{-1}$ for energy (Faber et al., 2017)), so any remaining difference to real-world data is almost exclusively due to errors in the DFT simulation. Truly reaching better accuracy can therefore only be achieved with more precise ground-truth data, which requires far more expensive methods (e.g. CCSD(T)) and thus ML models that are more sample-efficient (Chmiela et al., 2018). We therefore instead test our model on the harder task of using only 1000 training samples. As shown in Table 2 DimeNet outperforms SchNet by a large margin and performs roughly on par with sGDML. However, sGDML uses hand-engineered descriptors that provide a strong advantage for small datasets, can only be trained on a single molecule (a fixed set of atoms), and does not scale well with the number of atoms or training samples.

**Ablation studies.** To test whether directional message passing and the Fourier-Bessel basis are the actual reason for DimeNet's improved performance, we ablate them individually and compare the mean standardized MAE and logMAE for multi-task learning on QM9. Table 3 shows that both of our contributions have a significant impact on the model's performance. Using 64 Gaussian RBFs

instead of 16 and 6 Bessel basis functions to represent $d_{ji}$ and $d_{kj}$ increases the error by $10\,\%$, which shows that this basis does not only reduce the number of parameters but additionally provides a helpful inductive bias. DimeNet's error increases by around $26\,\%$ when we ignore the angles between messages by setting $N_{\mathrm{SHBF}} = 1$, showing that directly incorporating directional information does indeed improve performance. Using node embeddings instead of message embeddings (and hence also ignoring directional information) has the largest impact and increases MAE by $68\,\%$, at which point DimeNet performs worse than SchNet. Furthermore, Fig. 5 shows that the filters exhibit a structurally meaningful dependence on both the distance and angle. For example, some of these filters are clearly being activated by benzene rings ($120°$ angle, $1.39\,\text{Å}$ distance). This further demonstrates that the model learns to leverage directional information.

## 8 CONCLUSION

In this work we have introduced directional message passing, a more powerful and expressive interaction scheme for molecular predictions. Directional message passing enables graph neural networks to leverage directional information in addition to the interatomic distances that are used by normal GNNs. We have shown that interatomic distances can be represented in a principled and effective manner using spherical Bessel functions. We have furthermore shown that this representation can be extended to directional information by leveraging 2D spherical Fourier-Bessel basis functions. We have leveraged these innovations to construct DimeNet, a GNN suitable both for predicting molecular properties and for use in molecular dynamics simulations. We have demonstrated DimeNet's performance on QM9 and MD17 and shown that our contributions are the essential ingredients that enable DimeNet's state-of-the-art performance. DimeNet directly models the first two terms in Eq. 1, which are known as the important "hard" degrees of freedom in molecules (Leach, 2001). Future work should aim at also incorporating the third and fourth terms of this equation. This could improve predictions even further and enable the application to molecules much larger than those used in common benchmarks like QM9.

### ACKNOWLEDGMENTS

This research was supported by the German Federal Ministry of Education and Research (BMBF), grant no. 01IS18036B, and by the Deutsche Forschungsgemeinschaft (DFG) through the Emmy Noether grant GU 1409/2-1 and the TUM International Graduate School of Science and Engineering (IGSSE), GSC 81. The authors of this work take full responsibilities for its content.

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

## A    INDISTINGUISHABLE MOLECULES

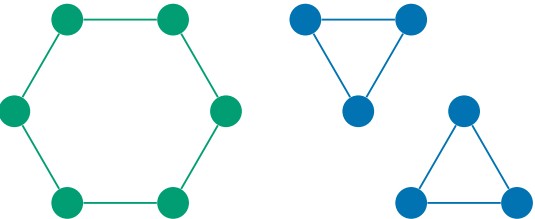

Figure 6: A standard non-directional GNN cannot distinguish between a hexagonal (left) and two triangular molecules (right) with the same bond lengths, since the neighborhood of each atom is exactly the same. An example of this would be Cyclohexane and two Cyclopropane molecules with slightly stretched bonds, when the GNN either uses the molecular graph or a cutoff distance of $c \leq 2.5$ Å. Directional message passing solves this problem by considering the direction of each bond.

## B    EXPERIMENTAL SETUP

The model architecture and hyperparameters were optimized using the QM9 validation set. We use 6 stacked interaction blocks and embeddings of size $F = 128$ throughout the model. For the basis functions we choose $N_{\text{SHBF}} = 7$ and $N_{\text{SRBF}} = N_{\text{RBF}} = 6$. For the weight tensor in the interaction block we use $N_{\text{bilinear}} = 8$. We did not find the model to be very sensitive to these values as long as they were large enough (i.e. at least 4).

We found the cutoff $c = 5$ Å and the learning rate $1 \times 10^{-3}$ to be rather important hyperparameters. We optimized the model using AMSGrad (Reddi et al., 2018) with 32 molecules per mini-batch. We use a linear learning rate warm-up over 3000 steps and an exponential decay with ratio 0.1 every $2\,000\,000$ steps. The model weights for validation and test were obtained using an exponential moving average (EMA) with decay rate 0.999. For MD17 we use the loss function from Eq. 2 with force weight $\rho = 100$, like previous models Schütt et al. (2017). Note that $\rho$ presents a trade-off between energy and force accuracy. It should be chosen rather high since the forces determine the dynamics of the chemical system (Unke & Meuwly, 2019). We use early stopping on the validation loss. On QM9 we train for at most $3\,000\,000$ and on MD17 for at most $100\,000$ steps.

## C    SUMMARY STATISTICS

We summarize the results across different targets using the mean standardized MAE

$$\text{std. MAE} = \frac{1}{M} \sum_{m=1}^{M} \left( \frac{1}{N} \sum_{i=1}^{N} \frac{|f_{\theta}^{(m)}(\boldsymbol{X}_i, \boldsymbol{z}_i) - \hat{t}_i^{(m)}|}{\sigma_m} \right), \tag{10}$$

and the mean standardized logMAE

$$\text{logMAE} = \frac{1}{M} \sum_{m=1}^{M} \log \left( \frac{1}{N} \sum_{i=1}^{N} \frac{|f_{\theta}^{(m)}(\boldsymbol{X}_i, \boldsymbol{z}_i) - \hat{t}_i^{(m)}|}{\sigma_m} \right), \tag{11}$$

with target index $m$, number of targets $M = 12$, dataset size $N$, ground truth values $\hat{\boldsymbol{t}}^{(m)}$, model $f_{\theta}^{(m)}$, inputs $\boldsymbol{X}_i$ and $\boldsymbol{z}_i$, and standard deviation $\sigma_m$ of $\hat{\boldsymbol{t}}^{(m)}$. Std. MAE reflects the average error compared to the standard deviation of each target. Since this error is dominated by a few difficult targets (e.g. $\epsilon_{\text{HOMO}}$) we also report logMAE, which reflects every relative improvement equally but is sensitive to outliers, such as SchNet's result on $\langle R^2 \rangle$.

# D   DIMENET FILTERS

To illustrate the filters learned by DimeNet we separate the spatial dependency in the interaction function $f_{\text{int}}$ via

$$f_{\text{int}}(\boldsymbol{m}, d_{ji}, d_{kj}, \alpha_{(kj,ji)}) = \sum_n [\sigma(\boldsymbol{Wm} + \boldsymbol{b})]_n \, f_{\text{filter1},n}(d_{ji}) f_{\text{filter2},n}(d_{kj}, \alpha_{(kj,ji)}). \qquad (12)$$

The filters $f_{\text{filter1},n} : \mathbb{R}^+ \to \mathbb{R}$ and $f_{\text{filter2},n} : \mathbb{R}^+ \times [0, 2\pi] \to \mathbb{R}^F$ are given by

$$f_{\text{filter1},n}(d) = (\boldsymbol{W}_{\text{RBF}}\boldsymbol{e}_{\text{RBF}}(d))_n, \qquad (13)$$

$$f_{\text{filter2},n}(d, \alpha) = (\boldsymbol{W}_{\text{SBF}}\boldsymbol{a}_{\text{SBF}}(d, \alpha))^T \mathbf{W}_n, \qquad (14)$$

where $\boldsymbol{W}_{\text{RBF}}$, $\boldsymbol{W}_{\text{SBF}}$, and $\mathbf{W}$ are learned weight matrices/tensors, $\boldsymbol{e}_{\text{RBF}}(d)$ is the radial basis representation, and $\boldsymbol{a}_{\text{SBF}}(d, \alpha)$ is the 2D spherical Fourier-Bessel representation. Fig. 5 shows how the first 15 elements of $f_{\text{filter2},n}(d, \alpha)$ vary with $d$ and $\alpha$ when choosing the tensor slice $n = 1$ (with $\alpha = 0$ at the top of the figure).

# E   MULTI-TARGET RESULTS

Table 4: MAE on QM9 with multi-target learning. Single-target learning significantly improves performance on all targets. Using a separate output block per target slightly reduces this difference with little impact on training time.

| Target | Unit | Multi-target | Sep. output blocks | Single-target |
|---|---|---|---|---|
| $\mu$ | D | 0.0775 | 0.0815 | 0.0286 |
| $\alpha$ | $a_0{}^3$ | 0.0649 | 0.0616 | 0.0469 |
| $\epsilon_{\text{HOMO}}$ | meV | 45.1 | 45.5 | 27.8 |
| $\epsilon_{\text{LUMO}}$ | meV | 41.1 | 33.9 | 19.7 |
| $\Delta\epsilon$ | meV | 59.2 | 63.6 | 34.8 |
| $\langle R^2 \rangle$ | $a_0{}^2$ | 0.345 | 0.348 | 0.331 |
| ZPVE | meV | 2.87 | 1.44 | 1.29 |
| $U_0$ | meV | 12.9 | 10.6 | 8.02 |
| $U$ | meV | 13.0 | 10.5 | 7.89 |
| $H$ | meV | 13.0 | 10.4 | 8.11 |
| $G$ | meV | 13.8 | 10.8 | 8.98 |
| $c_{\text{v}}$ | $\frac{\text{cal}}{\text{mol K}}$ | 0.0309 | 0.0283 | 0.0249 |
| std. MAE | % | 1.92 | 1.90 | 1.05 |
| logMAE | - | $-5.07$ | $-5.21$ | $-5.57$ |

