# OpenReview forum: "Directional Message Passing for Molecular Graphs"
_ICLR.cc/2020/Conference — Accept (Spotlight)_

### Official Review · AnonReviewer2 · 2019-10-22
**Official Blind Review #2**

**Rating:** 8

**Review:**

This paper beneficially incorporates directional information into graph
neural networks for molecular modeling while preserving equivariance.

This paper is a tour de force of architecture engineering.  Continuous
equivariance, potential field representation, and bandwidth limited
basis functions are synthesized in a compelling manner.

I found the exposition extremely intelligible despite my lack of
familiarity with molecular modeling.

The contribution is clear, although applicability beyond the specific
domain of molecular modeling is possibly limited.  Being continuously
equivariant with respect to rotations is interesting, but seems to require
problems where the input is encoded as a point cloud in a vector space;
I'm not familiar with such problems.  Nonetheless, the domain of molecular
modeling is sufficiently important in isolation.

I recommend acceptance, because the contribution is strong, the writing
is excellent, the ideas are well-motivated, and the experiments support
the empirical claims.


**Experience Assessment:**

I do not know much about this area.

**Review Assessment: Checking Correctness Of Derivations And Theory:**

I assessed the sensibility of the derivations and theory.

**Review Assessment: Checking Correctness Of Experiments:**

I assessed the sensibility of the experiments.

**Review Assessment: Thoroughness In Paper Reading:**

I read the paper at least twice and used my best judgement in assessing the paper.

---

> ### Author Response · Authors · 2019-11-11
> **DimeNet beyond molecules**
>
> As you correctly pointed out, going beyond the graph and incorporating the underlying spatial data is one of the main ideas behind our model. In our work we focus on molecular prediction and leverage the characteristics of this problem. However, many of the ideas we propose in our paper should be applicable to other graphs with underlying geometric data as well, such as meshes in computer vision.

---

### Official Review · AnonReviewer3 · 2019-10-23
**Official Blind Review #3**

**Rating:** 8

**Review:**


This is a sophisticated paper on predicting molecular properties at atom as well as molecular levels using physics inspired, extended GNN architectures. Two key extensions are provided above and beyond previous GNN models that operated on graphs derived from pairwise distances between atoms. First, the encoding of atom distances for use in neural messages is no longer done in terms of Gaussian radial basis function representations but in terms of spherical Bessel functions. The provide an orthogonal decomposition at resolutions controlled by associated frequencies. The orthogonality is though lost due to the use of an envelop function to ensure differentiability at cutoff distance defining the graph (essential for molecular simulations) but this appears not to affect performance. The second and the primary contribution of the paper, beyond the architecture itself, is the use of directional embeddings of messages where angles are transformed into cosine basis for neural mappings. In other words, the message sent from atom i to j aggregates messages from i's other neighbors in a manner that takes into account the angle formed by i, j, and k's respective atom positions. Since local directions are equivariant with an overall molecular rotation, the message passing architecture in this new representation remains invariant to rotations/translations. The added directional information, embedded in local basis representation, clearly makes the network more powerful (able to distinguish higher order structures).

- the authors suggest that the radial information can be transformed simply by element-wise multiplication while angular information requires more complex transformations in message calculations. Is there a physical insight to this or is this simply an empirical finding?

- there are many layers of transformations introduced from the atom embeddings before reaching the property of interest. Are so many layers really necessary?

- it seems models for QM9 data were trained separately for each different physical target. Is this really necessary? Given the many layers of transformations until the properties are predicted, couldn't the message passing component be largely shared?

- what exactly is the training data for the molecular simulation tests? The description in the paper is insufficient. A separate model is trained for each molecule, presumably based on samples resulting from physical simulations (?). What is provided to the models based on each "sample"?

- the ablation studies are helpful to assess the impact of the main differences (directionality, bessel vs Gaussian, node embeddings vs message) though I would wish to see what the degradation effect is on QM9 if one used a shared message passing architecture (just sharing the messages, resulting embeddings could be transformed separately for different predictions).

There's a recent workshop paper also making use of directional information (local coordinate transformations along a protein backbone chain) in message passing/transformer architectures: Ingraham et al., Generative models for graph-based protein design, ICLR workshop 2019







**Experience Assessment:**

I have published in this field for several years.

**Review Assessment: Checking Correctness Of Derivations And Theory:**

I assessed the sensibility of the derivations and theory.

**Review Assessment: Checking Correctness Of Experiments:**

I assessed the sensibility of the experiments.

**Review Assessment: Thoroughness In Paper Reading:**

I read the paper at least twice and used my best judgement in assessing the paper.

---

> ### Author Response · Authors · 2019-11-11
> **Improved experiments**
>
> - During most of our model development process we used a single model to jointly predict all 12 targets. We only changed this for the final training runs. We have added the results of multi-target predictions in the updated appendix (both with shared and separate output blocks per target). Using a shared model increases std. MAE by 81%, even when using separate output blocks. Please note that competing models use single-target models as well.
>
> - We have extended the description of the MD17 dataset to incorporate your suggestions.
>
> - The finding that a more complex angular transformation helps while a more complex distance transformation does not is mainly empirical. An intuitive explanation for this might be the fact that a rotation in space is a more complex transformation than a translation (in terms of the required matrix operation).
>
> - We only take over the architectural complexity from the model we are extending upon (PhysNet) and simplified where it was possible without sacrificing performance.
>
> - We have added the suggested workshop paper to the related work section. Note, however, that their method of incorporating orientations requires an explicit order of atoms, which breaks permutation invariance and is therefore only possible for chain-like molecules like proteins.

---

### Official Review · AnonReviewer1 · 2019-10-23
**Official Blind Review #1**

**Rating:** 6

**Review:**

Strength:
-- The paper is well written and easy to follow
-- The authors proposed a new approach called directional message passing to model the angles between atoms, which is missing in existing graph neural networks for molecule representation learning
--  The proposed approach are effective on some targets.

Weakness:
-- From the point of view of graph neural networks, the novelty of the proposed techniques is marginal
-- The performance of the proposed method are only better than existing methods on some of the targets.

This paper studied learning the graph representation of molecules by considering the angles between atoms. The authors proposed a specific type of graph neural network called directional message passing. Experimental results on the QM9 data set prove the effectiveness of the proposed approach over existing sota algorithms such as Schnet for some of the targets.

Overall, the paper studies a very important problem, which aims to learn the representation of molecules. Modeling the angles of atoms is indeed less explored in existing literature. From the view of graph neural networks, the proposed technique is not that new since edge embedding has already been studied in existing literature. But for the technique could be particular useful for molecule representation learning, especially with the BESSEL FUNCTIONS. One question is that the schnet also leverages the coordinates of the atoms, which may also implicitly model the angles between the edges. In terms of the experiments, the proposed approach does not seem that strong, only achieving the best performance on 5 targets out of 12.

Overall, I feel this paper is on the borderline. Now I will give weak accept and will revise my score according to other reviews and comments.

**Experience Assessment:**

I have published one or two papers in this area.

**Review Assessment: Checking Correctness Of Derivations And Theory:**

I assessed the sensibility of the derivations and theory.

**Review Assessment: Checking Correctness Of Experiments:**

I carefully checked the experiments.

**Review Assessment: Thoroughness In Paper Reading:**

I read the paper at least twice and used my best judgement in assessing the paper.

---

> ### Author Response · Authors · 2019-11-11
> **Performance improvements and directionality**
>
> Performance improvements: After the submission deadline we noticed that competing models use the raw (non-standardized) data for training. To make the setup more consistent we changed this training detail, which made a surprisingly large difference. We achieved further small improvements by jointly representing interatomic distances and angles in a 2D basis and by increasing the number of interaction blocks from 4 to 6. We’ve updated the description in the paper accordingly (see Sec. 5). We now set the state of the art on 11 out of 12 targets.
>
> Directionality has not been explored in GNNs: Our proposed directional message embeddings are indeed related to edge embeddings, as we point out in the related work section. However, normal edge embeddings do not incorporate any directional information. By using directional information we essentially go beyond the graph representation and leverage the fact that molecules are in fact embedded in 3D space. Also, note that our model outcompetes a state-of-the-art GNN with edge embeddings (MEGNet, published in April 2019) on average by 71%, while relying solely on message embeddings. MEGNet on the other hand uses a combination of node, edge and graph embeddings.
>
> Implicit angles: Since SchNet only uses the distances between atoms it can not model angles (even implicitly) when it only passes messages between one-hop neighbors (see e.g. limitations of the Weisfeiler-Lehman test and our discussion in Section 4 and Appendix A). Models like SchNet can only use the full geometrical information when they use the full distance matrix, i.e. a fully connected graph and no cutoff.

---

### Decision · Program_Chairs · 2019-12-19

**Decision:**

Accept (Spotlight)

**Comment:**

This paper studies Graph Neural Networks for quantum chemistry by incorporating a number of physics-informed innovations into the architecture. In particular, it considers directional edge information while preserving equivariance.

Reviewers were in agreement that this is an excellent paper with strong empirical results, great empirical evaluation and clear exposition. Despite some concerns about the limited novelty in terms of GNN methodology ( for instance, directional message passing has appeared in previous GNN papers, see e.g. https://openreview.net/forum?id=H1g0Z3A9Fm , in a different context). Ultimately, the AC believes this is a strong, high quality work that will be of broad interest, and thus recommends acceptance.